# Biochemical and Metabolical Pathways Associated with Microbiota-Derived Butyrate in Colorectal Cancer and Omega-3 Fatty Acids Implications: A Narrative Review

**DOI:** 10.3390/nu14061152

**Published:** 2022-03-09

**Authors:** Adelina Silvana Gheorghe, Șerban Mircea Negru, Mădălina Preda, Raluca Ioana Mihăilă, Isabela Anda Komporaly, Elena Adriana Dumitrescu, Cristian Virgil Lungulescu, Lidia Anca Kajanto, Bogdan Georgescu, Emanuel Alin Radu, Dana Lucia Stănculeanu

**Affiliations:** 1Department of Oncology, “Carol Davila” University of Medicine and Pharmacy, 020021 Bucharest, Romania; adelina-silvana.gheorghe@drd.umfcd.ro (A.S.G.); raluca-ioana.mihaila@drd.umfcd.ro (R.I.M.); isabela-anda.komporaly@drd.umfcd.ro (I.A.K.); elena-adriana.dumitrescu@drd.umfcd.ro (E.A.D.); lidia.kajanto@drd.umfcd.ro (L.A.K.); bogdan.georgescu@drd.umfcd.ro (B.G.); alin.radu@umfcd.ro (E.A.R.); dana.stanculeanu@umfcd.ro (D.L.S.); 2Department of Oncology, “Victor Babeș” University of Medicine and Pharmacy, 300041 Timisoara, Romania; 3Department of Microbiology, Parasitology and Virology, Faculty of Midwives and Nursing, “Carol Davila” University of Medicine and Pharmacy, 020021 Bucharest, Romania; 4Department of Medical Oncology, University of Medicine and Pharmacy Craiova, 200349 Craiova, Romania; cristilungulescu@yahoo.com

**Keywords:** butyrate, colorectal cancer, gut microbiota, diet, omega-3 PUFAs

## Abstract

Knowledge regarding the influence of the microbial community in cancer promotion or protection has expanded even more through the study of bacterial metabolic products and how they can modulate cancer risk, which represents an extremely challenging approach for the relationship between intestinal microbiota and colorectal cancer (CRC). This review discusses research progress on the effect of bacterial dysbiosis from a metabolic point of view, particularly on the biochemical mechanisms of butyrate, one of the main short chain fatty acids (SCFAs) with anti-inflammatory and anti-tumor properties in CRC. Increased daily intake of omega-3 polyunsaturated fatty acids (PUFAs) significantly increases the density of bacteria that are known to produce butyrate. Omega-3 PUFAs have been proposed as a treatment to prevent gut microbiota dysregulation and lower the risk or progression of CRC.

## 1. Introduction

According to the estimates of cancer incidence and mortality produced by the International Agency for Research on Cancer (GLOBOCAN) for 2020, colorectal cancer (CRC) represents one of the major health threats, especially in developed countries which are at a higher risk, due to the dietary and lifestyle patterns of the population [1,2]. It is the second leading cause of cancer death worldwide and the third most diagnosed cancer, while having a remarkable geographical variation [2,3]. As the global burden of CRC is expected to increase by 60% until 2030, with incidence and mortality rates rapidly rising in many countries in direct correlation with economic development and environmental changes, targeted interventions are necessary to reduce the number of patients in future years [4].

Complex genetic and epigenetic alterations contribute to the heterogeneity of CRC, having a multifactorial etiology and a wide variety of risk factors [3,5]. Besides the chromosomal and molecular characteristics involved in tumors arising from the colon and rectum, which have genetic signatures belonging to three major pathways (chromosomal instability; mismatch repair; and CpG island methylator phenotype), outstanding evidence based on global scientific and epidemiological studies emphasizes the role of environmental or lifestyle modifiable risk factors, such as a sedentary lifestyle or physical inactivity, obesity, smoking, processed food, alcohol, and meat consumption [3,5]. The human microbiome has a well-established impact on human physiology and disease, emerging as one of the factors involved in oncogenesis in the last few years. There is strong evidence indicating how gut microbiota can influence the development of CRC through multi-step disturbances of composition or function [6]. Recent advances and accumulating evidence of pathogens’ pro-carcinogenic effects, by promoting chronic inflammation, DNA alteration and activation of anti-apoptotic signaling pathways in malignant cells have contributed to understanding the etiology of several types of cancer, including CRC [7,8]. Pathogenic bacteria contribute to the pro-tumorigenic microenvironment through different mechanisms, such as outcompeting commensals, altering pH, secreting toxins or possessing virulence factors that can disrupt the epithelial barrier and activate the inflammation cascade [7].

The first consensus led by a group of 18 experts on the cancer-associated microbiome had been published in 2019. More than half (10) agreed that there is a causal relationship between the human microbiome and the etiopathogenesis of some cancers [9]. The consensus acknowledged the lacks in the existing evidence and the need for future research in the field. The conclusions were that the microbiome, the environmental factors and an epigenetically or genetically vulnerable host, represents one of the three parts of the multidirectional “interactome”/“interome” leading to carcinogenesis [9].

Omega-3 polyunsaturated fatty acids (omega-3-PUFAs) are generally recognized both as dietary supplements and prescription medications. Increasing data from epidemiologic, clinical, and preclinical investigations show that omega-3-PUFAs are effective in preventing CRC [10].

## 2. The Gut Microbiome and Colorectal Cancer

The 20th century has brought an incomparable revolution in understanding the microbiota (formed by commensal symbionts, pathogens, or pathobionts) and their association to more than 100 essentially different disorders [11]. A comprehensive review, in which each microorganism was associated with various diseases, local or external factors and antibiotic consumption, has recently outlined the global picture of the human microbiota composition (community of microbes residing in and on the human body) and function of the microbiome, defined as the combined genetic material of the microbiota [12].

There was a long process that led to current insights, as the first descriptions of the human microbiota date back to the 1680s, when Antonie van Leeuwenhoek began to use the microscope invented by himself to describe and illustrate his own bacteria in the swabs from his oral mucosa and faeces [11]. Almost 200 years later, the book published by Joseph Leidy can be considered as the “birth” of microbiota research. It was followed by the work of Pasteur, Metchnikoff, Koch, Escherich, Kendall and many others, who laid the groundwork for the present understanding of microorganism–host interactions [11]. The third millennium can be considered so far as the golden age of microbiota research, as it brought extensive population-based studies, such as the Human Microbiome Project (started five years after the completion of the Human Genome Project) or the American Gut Project.

Millions of years of co-evolution between host and microorganisms led to the formation of a mutualistic relationship, a complex interplay between the host’s immune system and the microbiota, essential for gut homeostasis [13]. In this interaction, microbiota’s main contributions to the host are the digestion and fermentation of carbohydrates, the production of vitamins, the development of gut-associated lymphoid tissues, the polarization of gut-specific immune responses, and prevention of colonization by pathogens. In contrast, the gut immune responses induced by commensal bacteria can regulate the composition of the microbiota [14]. There is also a bidirectional relationship between bacteria and metabolites, with bacteria influencing metabolite composition of the gut and metabolites contributing to the architecture of microbiota [15].

Molecular approaches through metagenomics (sequencing of all the DNA present in a sample) have firstly characterized the genetic background (gene composition) of the microbiome, offering an in-depth understanding of the complex and diverse bacterial communities [16]. However, the need for additional analysis of other datasets has rapidly emerged, as more knowledge was required about the host–microorganisms interplay, because metagenomic assays could not capture some bacterial features, they could not detect minority populations, and could not discriminate between live bacteria and transient DNA [13]. Therefore, researchers have used broad metagenomics technology integrated with metagenomic data, such as metatranscriptomics (a sequencing of all the mRNA generated to determine gene expression); culturomics (a culturing approach using multiple culture conditions; MALDI-TOF mass spectrometry and 16S rRNA sequencing); metaproteomics (analysis of microbial proteins that are actively synthesised by the microbiota): toxicogenomics (DNA-DNA hybridization, DNA G + C content and 16S rRNA sequence similarity); and metabolomics (the study of all chemical processes concerning microbiota’s metabolites) [12,17,18].

Studying the metabolic pathway information of the gut microbiota is particularly unique, as metabolites are considered to be universal, while proteins and genes vary across taxa. More research should be focused on gaining knowledge of the important mutual dependence between mucosal gut-associated metabolome composition and the gut microbiome, manifested by two general processes: the first one represented by catabolism and anabolism of metabolites by microbes, and the second by stimulation and inhibition of microbial growth by metabolites [19]. This “interomic” integrative analysis is required in order to transform the metabolites into a specific target for diagnosing or monitoring CRC or other microbiome-associated intestinal diseases and provide a potential therapeutic intervention (either directly, or indirectly, through diet) [19].

To understand microbiota’s metabolic transformation capabilities and how they can affect the host is essential to study the knowledge gap in the molecular basis for gut microbiota–nutrient interactions (nutritional requirements, links between diet and disease, the effect of diet on microbiota) also from a biochemical point of view, in strong correlation with host nutrition [20]. The altered microbiota can lead to a disruption in the mucosal barrier, promote or inhibit tumorigenesis through different immune responses and microbiome-derived metabolites [21]. On the one hand, there are some microbial metabolites, such as prostaglandin E2 and secondary bile acids, associated with increased risk of CRC and on the other one, some are associated with decreased CRC risk: indole; antioxidants; short-chain fatty acids (SCFAs); and ursodeoxycholic acid [22,23].

A chronic state of unhealthy and dysbiotic-acquired microbiota, defined as aberrant composition and function, with a high resilience potential, has been shown to contribute to cancer pathogenesis, both directly and indirectly via: prolonged inflammation; promotion of cell growth and proliferation; changes in immune responses (lessening the strength of immunosurveillance); metabolic changes (alteration of food and drug metabolism or other biochemical functions of the host); and DNA damage and alterations of the anti-cancer therapy efficacy [14,24]. Microbial dysbiosis has intricate connections with neoplastic diseases, especially with CRC, as the most important and developed community of human microbiota resides in the gastrointestinal tract.

However, recent research has also identified several mechanisms through which gut microbiota may support the host’s fight against cancer, such as the use of antigenic mimicry, biotransformation of chemotherapeutic agents, boosting of anti-cancer immune responses (to improve the efficacy of cancer immunotherapy) and producing microbial metabolites with tumor suppressing properties [24].

In addition to the laboratory research, there are many investigations currently taking place also in the clinical setting. Prospective cohort studies of patients with CRC could distinguish early-stage patients from more advanced disease, based on the gut microbiome and metabolome. For example, a clinical trial (NCT04005742) with unpublished results yet, aims to perform additional measurements to the ones in the BORICC study (Biomarkers of Risk of Colorectal Cancer), including the gut microbiome, fecal SCFAs concentrations and expression of genes associated with CRC, in a 12+ year follow-up longitudinal study (BFU—BORICC Follow-Up) [25]. The ultimate goal of the BFU study is to identify lifestyle factors able to reduce CRC risk while characterizing the underlying mechanisms in which lifestyle and aging affect CRC risk to design better early prevention strategies [25].

## 3. Microbiota-Derived Butyrate in Colorectal Cancer

SCFAs are weak organic acids with between two and five carbon molecules, including acetate (C2); propionate (C3); butyrate (C4); and valerate (C5), which are produced by the intestinal bacterial fermentation of mainly undigested dietary carbohydrates (especially resistant starches and dietary fibers) [26]. However, they also result in small quantities from dietary and endogenous proteins, through a pathway that also produces toxic nitrogenous and sulfur metabolites, such as ammonia (from the conversion of the amino acid lysine into butyrate) [27]. The concentration ratio in the colonic lumen of the three main SCFAs is about 3/1/1: 60% acetate, 25% propionate, and 15% butyrate, the last one being the preferred energy source utilized by epithelial cells from the colon, and only small proportions reach the portal vein or the systemic circulation [26].

In vitro studies have demonstrated the important role of butyric acid in the prevention of CRC. An evaluation of HCT116 human CRC cells treated with butyric acid derivatives proved that apoptosis is induced in the cancer cells by activation of caspase-3 activity and induced cell cycle arrest [28]. Moreover, SCFAs can also regulate the expression of inflammatory cytokines and chemokines by the colonic epithelial cells in different immune processes, having a role in gut homeostasis and promoting the integrity of the intestinal barrier [29]. The effect of butyrate on the epithelial integrity has been confirmed also in animal models, where it contributes to the healing of colonic tissue at the anastomosis sites after surgery for CRC [30].

Butyrate-producing bacteria are an abundant and phylogenetically diverse group of microorganisms, considered to be a functional group of Gram-positive anaerobic Firmicutes, which play an important role in maintaining a healthy gut, primarily through their production of butyrate [26,31]. Two of the most numerically important groups are considered to be *Faecalibacterium prausnitzii*, belonging to the *Clostridium leptum* cluster (clostridial cluster IV) and *Eubacterium rectale*/*Roseburia* spp., belonging to the *Clostridium coccoides* cluster (clostridial cluster XIVa) [31]. There are two microbial enzymes responsible for the final synthesis of butyrate from two molecules of acetyl-CoA: butyryl-CoA transferase (dominant, formed by a variety of genera and species) and butyrate kinase (favored in proteolytic fermentation) [6].

Increased butyrate production has often been hypothesized to be one of the beneficial effects of prebiotics and probiotics [29]. One important prebiotic is represented by the resistant starch (the starch which escapes the digestion in the small intestine), which reduced colonic neoplasia in studies including carcinogen-treated rats but increased intestinal tumorigenesis in the genetically driven Apc1638N mouse model [32]. The antineoplastic effect of resistant starch can be due to fermentation end-products, mainly butyrate [32]. A mouse model study has shown that in the ones colonized with butyrate-producing bacterium the high fiber diet had a protective role, but not in the mice lacking a butyrate-producing bacterium [33]. The same study also evaluated the protective effect in the case of mice colonized with a mutant strain of the butyrate-producing bacterium, harboring a deletion in the butyryl CoA synthesis operon which produces diminished levels of butyrate; the fiber diet had an attenuated protective effect with an intermediate tumor burden [33]. Another way by which resistant starch, together with other insoluble fiber, may prevent the colonic neoplasm is by speeding the colonic transit, thus reducing the exposure of epithelial cells to ingested carcinogens [34].

In a clinical trial (NCT03072641) which aimed to determine whether probiotic bacteria have a beneficial effect on the CRC-associated microbiota, researchers used dietary supplementation with *Bifidobacterium lactis* Bl-04 and *Lactobacillus acidophilus* NCFM and analyzed the microbiota composition in tissue and faeces samples, at baseline and after probiotics use [35]. The results showed that patients with CRC who received probiotics had an increased abundance of butyrate-producing bacteria (especially *Faecalibacterium* spp. and *Clostridiales* spp.) in the samples from the tumor, non-tumor mucosa, and faeces, compared with the group that did not receive probiotics. [35] Moreover, CRC-associated taxa (*Fusobacterium* and *Peptostreptococcus*) were less frequent in the fecal samples of patients who received probiotics, upholding the hypothesis that CRC-associated microbiota can be manipulated by specific probiotic strains and providing hope that the probiotics modulation of microbiota could be considered an integrative part of the therapeutic approach for CRC patients [35].

The structure of microbiota in patients with CRC was described as significantly different from the one encountered in healthy individuals. There are several studies of butyryl-CoA: acetate CoA-transferase gene quantification from the gut microbial population and they all found that butyrate-producing bacteria (such as *Ruminococcus* spp. and *Pseudobutyrivibrio ruminis*) are reduced in the feces of CRC patients, pointing out the benefits of bacterial metabolites [22,36,37]. An evaluation of diet and age suggested that these factors also influence the level of butyrate-producing bacteria in the gut—the older participants had significantly fewer copies of the butyryl-CoA:acetate CoA-transferase gene than young omnivores, while vegetarians showed the highest number [38]. This fact may reflect the increased risk for CRC in the elderly, due to their low butyrate production capacity and the protective effect of a vegetarian diet against CRC.

In a study that aimed to describe how microbial functions may influence CRC development, researchers used stool profiling to identify intestinal microbiome and metabolome and analyze the different representation in humans with CRC, compared to healthy controls [22]. They quantified several SCFAs from frozen stool samples, among which acetic and valeric acids were significantly higher in the feces from CRC patients. In contrast, butyric acid was significantly higher in the healthy samples, and propionic acid was detected in similar quantities between the two groups [22]. Acetate can be turned into butyrate, but the proportional differences in these two SCFAs metabolites between CRC and healthy individuals may be explained by a reduction of gut bacteria that can perform this reaction in CRC samples. Otherwise, it may be a result of the conversion of butyrate into acetate, a degradation process that takes place under the acidic (low) colonic pH induced by the tumor [22]. In CRC samples, significantly higher relative concentrations of isobutyric and isovaleric acid were observed as well, being products from the bacterial metabolism of branched-chain amino acids valine and leucine, also higher in CRC stool samples [22]. Butyrate proved to be more than just a metabolite, having important cellular signaling roles as well, linked to epigenetic regulations of gene expression. Butyrate can regulate the expression of a large number of genes by direct interaction with transcription factors such as p53; retinoblastoma protein; Stat3; NF-kB; and estrogen receptors, which are critical epigenetic regulators and a new class of anticancer agents [39]. Butyrate also has intracellular roles, like DNA methylation; histone methylation; hyperacetylation of nonhistone proteins; inhibition of histone phosphorylation; regulation of expression of micro-RNAs; and modulation of intracellular kinase signaling [39]. Moreover, butyrate can act as agonist of a G-protein-coupled receptor found in the apical membrane of human colonic epithelial cells, GPR109A [39].

There are two main forms of epigenetic changes encountered in CRC (and many other cancers), defined as chemical alterations to DNA or chromatin that do not affect the primary DNA sequence: those that directly modify DNA (DNA hypo- or hypermethylation) and those that modify DNA-binding proteins (histone modifications—methylation or demethylation, and the acetylation or deacetylation) [40]. These changes alter the regulation and expression of genes and other DNA elements in a predictable fashion and are reversible, unlike changes to the genomic sequence [40,41]. The enzymes that catalyze histone acetylation are called histone acetyltransferases (HATs) and the ones that catalyze the removal of an acetyl group from a histone are called histone deacetylases (HDACs), both playing a crucial role in the remodeling of chromatin [41]. Their enzymatic activities induce structural alterations of histones, enabling access of transcription factors to a portion of DNA chromatin, influencing the transcription and expression of a given gene [41].

Butyrate was the first identified endogenous inhibitor of HDAC, in 1977 and for more than two decades thereafter it was the only one available for research, with its primary target in the clinical development of cancer treatment [42]. Butyrate acts as an inhibitor of HDAC and leads to hyperacetylation of histones [39]. During the last decade, inhibition of HDACs by HDAC inhibitors (HDACIs) emerged as a target for specific epigenetic modification associated with cancer or other diseases (hemoglobinopathies; cystic fibrosis; X-linked adrenoleukodystrophy; muscular dystrophies; neurodegenerative disorders; systemic lupus erythematosus etc.). More than 20 substances have entered clinical studies by now, while some have already been approved (for example vorinostat orSuberAniloHydroxamic acid and romidepsin or depsipeptide for the treatment of cutaneous T-cell lymphoma or the drug panobinostat for the treatment of multiple myeloma) [43].

Due to the fact that HDACs are key enzymes for regulating cell death and have a role in promoting carcinogenesis, HDACIs have been exploited for their role in cancer therapy and have been shown to regulate the survival of tumor-infiltrating T lymphocytes (TILs), by suppressing their apoptosis [44]. HDACIs, particularly butyrate, also inhibit directly colon carcinogenesis, by decreasing the expression of cyclin B1 gene (a cell cycle promoter) in colon cancers cells, as shown in vitro [45]. Moreover, co-administration of HDACIs and anti-CTLA4 (cytotoxic T-lymphocyte antigen 4) antibodies seems to act synergistically for the therapeutic effect, enhancing T-cell infiltration within the tumor and the anti-tumor immune response [44]. Another mechanism by which microbiota-derived butyrate promotes cellular metabolism is the enhancement of memory potential in activated CD8^+^ T cells (through increased oxidative and glycolytic activity, improved mitochondrial mass and membrane potential), with implications in immunotherapy and vaccination [46].

Belcheva et al. conducted a study on the murine model that investigated the ability of gut microbiota to synergize with mutations in oncogenes or tumor suppressor genes (APC, MSH), as well as with established lifestyle risk factors related to diet (high-carbohydrate intake formed by starch and sucrose), has demonstrated the role of microbial-derived butyrate in the straightforward connection between host genetics and gut microbes [47]. Some of the most common genetic changes involved in this pathology are the mutation in or silencing of the genes involved in DNA mismatch repairs (MMR) mechanisms, such as *MutS homolog 2* (MSH2) and the mutation of the adenomatous polyposis coli (APC) tumor suppressor gene, which regulates the Wnt/*β*-catenin pathway [48]. There is important research concern over carbohydrate-rich diets; for example, a meta-analysis of 39 studies suggested an overall direct association between glucose metabolism factors (glycemic index, glycemic load) and CRC risk [49]. In a prospective, observational study of participants from the National Cancer Institute sponsored Cancer and Leukemia Group B (CALGB) 89,803 trial (ClinicalTrials.gov identifier NCT00003835), regarding patients with stage III colon cancer, it had been stated that total carbohydrate intake and increased dietary glycemic load were significantly statistically associated with a higher risk of recurrence and mortality [50].

The mentioned study of Belcheva et al. was conducted on APC^Min/+^ (multiple intestinal neoplasia) mouse, a well-established animal model of human adenomatous polyposis, with MSH2 deficiency: an APC^Min/+^MSH2^−/−^ mouse model of CRC [47]. Alteration of the gut bacterial community structure with antibiotics led to decreased polyp numbers in the mice colons, without a reduction in the abundance of colonic bacteria, through a mechanism independent of both inflammation and DNA damage [47]. Putting mice on a 7% low-carbohydrate diet led to substantial changes in the relative proportions of the bacterial phyla but did not alter the total bacterial abundance. Moreover, this diet reduced the number of polyps in the digestive tracts of APC^Min/+^MSH2^−/−^ mice, in a similar amount achieved by treating mice with antibiotics [47]. The antibiotic treatment and the low-carbohydrate diet had also numerous other effects observed in APC^Min/+^MSH2^−/−^ mice: both interventions reduced the number of cells with DNA breaks, reduced Ki-67 expression, modulated and restored the nuclear β-catenin expression to that encountered in APC^Min/+^MSH2^+/−^ mice, reduced the production of numerous metabolites (lactate, only butyrate from all SCFAs, uracil, xanthine etc.) of microbial fermentation, decreased three butyrate-producing families within the *Firmicutes* phylum (*Clostridiaceae*, *Lachnospiraceae* and *Ruminococcaceae*) without impacting total bacterial abundance, and reduced the gene copy number for butyryl-CoA transferase [47]. All these results support the idea that the gut microbiota-related metabolome plays a crucial role in CRC by providing metabolites such as butyrate.

Histopathological studies have shown that adenomatous polyps appear through a top-down morphogenesis mechanism, from the apex to the bottom of the crypts, as the more time cells reside in the mucosa, the more chances exist for epigenetic alterations through carcinogens exposure, required for tumor formation [51]. This process is in contradiction with the generally accepted statement that cancer cells derive from normal stem cells, as in the colon they exist near the base of the crypts. Therefore, in the superficial parts of the crypts, there are dysplastic epithelial cells which present a markedly abnormal pattern of Ki-67 proliferation and genetic alterations of APC locus (loss of heterozygosity) leading to functional changes in β-catenin expression and localization, these mutant clones being genetically unrelated to the cells from the bottom of the crypt [51]. In the light of this particularity of the intestinal epithelium, it has been hypothesized that by influencing cell movement, profound effects on tumorigenesis may be obtained, since high-velocity cell loss represents an efficient way of eliminating cells that have acquired mutations and preventing irreversible cancerous phenotype by longer exposure to carcinogens [52].

A study on murine APC^+/Min^ epithelial cells showed that in vitro, they are less motile than APC^+/+^ cells and possess a disarranged actin cytoskeletal network, properties which make them more prone to acquiring additional genetic alterations and forming tumors [52]. Treatment with two mM butyrate for 24 h was demonstrated to increase haptotaxis in both cellular lines, acting as a promoter of the migration of colonic cancerous epithelial cells. The effect was greater in the APC^+/Min^ cell line, as it was able to restore both motile function and actin cytoskeletal organization seen in APC^+/+^ cells [52]. The link between butyrate treatment and cytoskeleton assembly can be explained by its capacity for protein acetylation, which has a key role in these fibers’ function. Moreover, exposure to high concentrations of butyrate (5 mM) induced apoptosis in the mutated cells, measured by caspase-3-like activity [52]. These results may explain the protective effect determined by butyrogenic diets on CRC carcinogenesis, by increasing colonocyte velocity and shortening the exposure of cells to carcinogens, especially in the cases with APC or the β-catenin gene mutations.

Another study that assessed butyrate’s effect on the motility of colonocytes investigated its ability to act directly at the molecular level of the cytoskeletal components from ileal and colonic smooth muscle cells in primary culture and on A7R5 murine cell line [48]. It was shown that butyrate (>0.1 mM) inhibited myocytes’ proliferation in the A7R5 line. This finding also applied in primary culture, but only at higher concentrations, while butyrate in low concentration (0.05–0.5 mM) significantly stimulated the proliferation of myocytes [53]. Other observed effects of butyrate included the stimulation of collagenous’ and noncollagenous’ protein synthesis, as well as enhancement of actin and myosin expression [53]. Butyrate’s activity on the contractibility of colonic smooth muscle proves to be dependent on its concentration in the lumen, besides the intracellular butyrate concentration (dependent on the level of its oxidation).

The association between fecal SCFAs concentrations and the efficacy of immunotherapy may emerge as a new biomarker to monitor patients undergoing treatment with programmed cell death-1 (PD-1) inhibitors. Nomura et al. recently evaluated 52 patients with solid tumors treated with nivolumab or pembrolizumab and concluded that those with higher concentrations of fecal SCFAs had a longer progression-free survival and also response to anti-PD-1 therapy [54].

Some omega-3 fatty acids, like EPA and DHA, could have a potential adjuvant therapy role, thanks to their low toxicity profile and their capability to downregulate the expression of the efflux pump, P glycoprotein, in a doxorubicin-resistant variant of HT29 cells [55]. Multiple other studies have evaluated their potential as adjuvant agents in chemotherapy, for example the association of EPA and a regimen of 5-fluorouracil (5-FU) and oxaliplatin can have a synergistic anti-cancer effect [55]. Some authors concluded that adding omega-3 fatty acids in chemotherapy could restore lipid stocks and potentially limit 5-FU side effects [55].

Other studies have noticed an increased apoptosis of cancerous cells when adding EPA and DHA to 5-FU, oxaliplatin, and irinotecan [55]. Dietary intake of omega3 fatty acids and chemotherapy might have a synergetic effect [56]. Combined treatment of fish oil and 5-FU enhanced growth inhibition compared to cells exposed to either substance alone [56].

## 4. Role of Omega-3 Fatty Acids in Regulating Butyrate-Producing Gut Microbiota

Consumption of omega-3-PUFA-rich diets has been demonstrated to be beneficial for health, supporting a good quality of life and ameliorating or preventing several disorders (cardiovascular, inflammatory, neurodegenerative diseases, diabetes mellitus, and cancer) [57].

The effects of an omega-3-rich diet on gut microbiota were studied using animal models, proving there is a correlation between the two [58]. Dietary omega-3-PUFAs are largely digested in the distal intestine by anaerobic bacteria such as *Bifidobacteria* and *Lactobacilli*, influencing the intestinal flora distribution, being shown to improve gut microbial dysbiosis by increasing probiotic species and butyric acid-producing bacteria, according to several studies conducted on humans [59].

In a published case report, a healthy 45-year-old man who received 600 mg of omega-3 every day for 14 days had his feces sampled. Species diversity reduced after the intervention, although butyrate-producing bacteria increased. *Faecalibacterium prausnitzii* and *Akkermansia* spp. were found to be significantly reduced. There was found to be a remarkable increase in *Blautia*, a genus whose reduction is associated with increased risk of CRC. After the 14-day washout, alterations in the gut flora were reversed, implying that the gut microbiota is a living, dynamic ecosystem that is subject to dietary changes. Therefore, increases in butyrate-producing bacteria may be responsible for some of omega-3’s health advantages [60].

The increase in butyrate-producing bacteria may be influenced also by eicosapentaenoic acid (EPA) and docosahexaenoic acid (DHA), which together with prebiotic fermentable fibers may have protective effects against colonic neoplasm more due to increased apoptosis rather than decreased cell proliferation [61]. EPA and DHA increase *Lactobacillus* and reduces *Helicobacter* and *Fusobacterium nucleatum* [61]. The butyrate may be involved in colonocyte apoptosis through its effect to promote cellular oxidation, being able to produce cellular reactive oxygen species when metabolized [61]. EPA and DHA can be incorporated in cell membranes and are susceptible to oxidation thanks to their high degree of unsaturation [61]. Other direct roles of EPA and DHA on colorectal cancer cells include modulation of cyclooxygenase metabolism, alteration of lipid raft behavior, increase in lipid peroxidation, regulation of kinase pathways, induction of pro-apoptotic pathways, modulation of WNT/β-catenin pathway and others [55].

Besides their direct roles, EPA and DHA have an effect on cancer cells through their metabolites, like resolvins, docosatriens and maresins [62,63]. Resolvins (resolution phase interaction products), protectins, and maresins are endogenously generated from n-3 PUFAs [61]. Resolvins are bioactive compounds with potent anti-inflammatory and immunoregulatory actions, and anti-carcinogenic compounds [55,62,63].

Even though gut microbiota changes associated with omega-3-PUFAs are poorly understood, omega-3 fatty acids may aid in the treatment of colorectal cancer by increasing colon beneficial bacterial populations.

## 5. Conclusions and Future Perspectives

Given the important role of butyric acid in the prevention of CRC, therapies with exogenous SCFAs or prebiotic/probiotic administration to modulate bacterial metabolism in the gut are being proposed to reduce mucosal inflammation and induce apoptosis in cancer cells.

Omega-3-PUFAs may affect the balance of gut microorganisms, which may contribute to the occurrence and progression of CRC, particularly due to their ability to increase butyric acid-producing bacteria.

These discoveries may shed light on the mechanisms underlying omega-3-PUFAs’ impact on a variety of chronic conditions, as well as provide a framework for developing individualized medical treatments for CRC and other diseases. Supplementing the diet with omega-3 is likely to be a relevant potential mechanism for reducing CRC risk in a primary prevention setting, but it may also be appropriate for the possible use of omega-3-PUFAs as adjuvant treatment of CRC.

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
