# Peer review of "Biochemical and Metabolical Pathways Associated with Microbiota-Derived Butyrate in Colorectal Cancer and Omega-3 Fatty Acids Implications: A Narrative Review"

_nutrients, 2022, doi:10.3390/nu14061152_

Round 1

Reviewer 1 Report

Three issues need to be more fully described in this review:
•    If butyrate is critical to decrease the development of colorectal cancer, then the role of prebiotics, for example in resistant starch, as sources of butyrate must be more fully outlined other than passing comments in lines 188 and 369. Two references could be DOI: 10.1016/S1470-2045(12)70475-8 and DOI: 10.1002/mnfr.201500902. If this evidence is not convincing, then the authors may need to review their hypothesis. 
•    The section on omega3 fatty acids looks like an afterthought rather than an integral part of this proposal. What is the evidence that omega3 fatty acids, specifically EPA and DHA, increase butyrate concentrations in the colon and blood, rather than an increase in butyrate-forming bacteria? How important is this effect, compared to the many other effects of omega3 fatty acids? Do omega3 fatty acids only have an adjuvant role (PMCID: PMC6133177)? Are the anti-inflammatory metabolites of the omega3 fatty acids such as the resolvins and maresins more relevant than the dietary compounds themselves (DOI: 10.1007/s10555-018-9744-y) and more important than a potential increase in butyrate? Are omega3 fatty acids synergistic with chemotherapeutic interventions (DOI: 10.7762/cnr.2017.6.3.147) rather than butyrate generators? Thus, the omega3 section needs much more information.
•    Is there any direct evidence correlating an increased butyrate concentration with a decreased incidence of colorectal cancer? Is any relationship causal or casual?

Reviewer 2 Report

It was a pleasure to read this manuscript, and it seems that the authors did a great deal of work preparing it.
I have some suggestions to improve the quality of the paper.

First and most important: Authors should also add some information (in the new, separate section) about the relationship between omega-3 PUFAs and colorectal cancer and try to find specific mechanisms which link them.

The dots should be after the „]” in all section of the manuscript.
In my opinion, in the introduction, the authors should emphasize the aim of this paper. Please, add some (one sentence?) information on the interaction and interplay between o3 FA, microbiome and colorectal cancer, which will be helpful in the smooth transition to the specific aim of the following manuscript.
Line 131: please add "s" to acids or delate "s" from SCFAs
Line 151,338: add "s" to SCFA
Line 174: "observed" -  I recommend changing it to "confirmed"
Line 345: add "s" to PUFA

Round 2

Reviewer 1 Report

Appropriate changes have been made by the authors.